# Stakeholders' perspectives on capturing societal cost savings from a quality improvement initiative: A qualitative study

Daniëlle Kroon[1]*, Simone A. van Dulmen[1], Niek W. Stadhouders[1], Jonas Rosenstok[2], Baukje van den Heuvel[2], Gert P. Westert[1], Rudolf B. Kool[1], Patrick P. T. Jeurissen[1]

1 Department of IQ Health, Radboud Institute for Health Sciences, Radboud University Medical Center, Nijmegen, The Netherlands, 2 Department of Operating Rooms, Radboud University Medical Center, Nijmegen, The Netherlands

* danielle.kroon@radboudumc.nl

**Data Availability Statement:** Data cannot be shared publicly because the participants did not provide consent to share the transcripts to persons

## Abstract

### Background

Besides improving the quality of care, quality improvement initiatives often also intend to produce cost savings. An example is prehabilitation, which can reduce complication rates and the length of stay in the hospital. However, the process from utilization reductions to actual societal cost savings remains uncertain in practice. Our aim was to identify barriers and facilitators throughout this process. We used the implementation of prehabilitation in a Dutch hospital as a test case.

### Methods

We held 20 semi-structured interviews between June and November 2023. Eighteen stakeholders were affiliated with the hospital and two with different health insurers. Nine interviews were held face-to-face and 11 via Microsoft Teams. The interviews were recorded and transcribed. The first transcripts were inductively coded by two authors, the subsequent transcripts by one and checked by another. Differences were resolved through discussion.

### Results

We identified 20 barriers and 23 facilitators across four stages: reducing capacity, reducing departmental expenses, reducing hospital expenses and reducing insurer expenses. All participants expected that the excess capacity will be used for other priorities. This was perceived as highly valuable and as an efficiency gain. Other barriers to capture savings included the fear of losing resilience, flexibility, status and revenue. Misalignment between service contracts among hospitals and insurers can hinder the ability to financially incentivize cost reductions. Additionally, some contract types can hinder the transfer of hospital savings to insurers. Identified facilitators included shared savings agreements, an explicit strategy targeting all stages, and labor shortage, among others.

other than the researchers. In addition, the transcripts contain business-sensitive information and even anonymized raw data can be compromising. Data are available from the IQ health science department of the Radboudumc (contact via iqhealth@radboudumc.nl) for researchers who meet the criteria for access to confidential data.

**Funding:** The study was funded by The Dutch Ministry of Health, Welfare and Sport (grant number 331032). The funders had no role in study design, data collection and analysis, decision to publish, or preparation of the manuscript.

**Competing interests:** We would like to declare that two authors, Jonas Rosenstok and Baukje van den Heuvel, are prehabilitation program managers and they both participated as interviewee. This does not alter our adherence to PLOS ONE policies on sharing data and materials.

## Conclusion

This study systematically describes barriers and facilitators that prevent translating quality improvement initiatives into societal cost savings. Stakeholders expect that any saved capacity will be used for other priorities, including providing care due to the increasing demand. Capturing any cash savings does not occur automatically, emphasizing the need for a strategy targeting all stages.

## Introduction

Health expenditure growth is expected to outpace gross domestic product (GDP) growth in most member countries of the OECD (Organization for Economic Co-operation and Development) during this decade [1]. Policymakers and healthcare organizations are seeking effective methods to bend the cost curve while preserving or even improving the quality of care [2]. Although for the majority of the quality improvement initiatives the primary aim is enhancing quality of care, occasionally substantial cost-savings are estimated [3–7]. For example, discontinuing five low-value general surgery services in the United Kingdom could lead to an annual cost reduction of €150 million [8].

However, the translation of such theoretical savings of quality improvement initiatives into actual societal cash savings is complex and often not achieved [9]. This is challenging due to various reasons. For example, estimates of cost savings based on reimbursement prices overestimate true savings, because only variable costs, such as costs of disposable equipment and drugs, can be saved in short-term [10, 11]. One study found that these costs only cover 16% of total expenses in hospitals [12]. The majority of expenses, such as salary costs, purchasing costs of reusable medical devices and organizational overhead, are not directly saved when the volume of healthcare services is reduced. Moreover, since claims data do typically consist of cross-subsidies, the actual total costs may be either higher or lower than the official rates. Besides, the relation between external funding structures and internal allocation of resources is blurred [13].

While improving quality of care may free up hospital capacity through shorter hospital stays and reduction of diagnostic tests and procedures, the capacity may be refilled with new treatments. Due to existing incentives in many healthcare systems, such substitution with other care occurs automatically [9, 14–16]. A way to achieve cash savings is to actively discourage care substitution. This requires an investment of time and resources [17]. Excess capacity should be gradually reduced until it reaches a threshold for downsizing, i.e. capacity reductions must be sufficiently large to scale down one single nursing shift, medical specialist, ward, etc.

Because marginal revenues typically exceed marginal costs by far, in fee-for-service type payment systems, scaling down costs is unlikely to be sufficient to cover for losses in hospital revenues. Under a fixed budget revenues are protected but cost savings that are not passed through to payers may shoulder organizational slack and not be returned to society, for example by reductions in taxes or insurance premiums. Moreover, since healthcare costs naturally increase due to demographic changes, new technologies and other structural drivers, it is difficult to establish the accurate benchmark. Cost savings can generally be interpreted as a lower hospital growth rate rather than actual reductions in hospital costs. However, it is challenging deciding upon the appropriate benchmark to measure cost savings in terms of expected growth, historical growth or comparator hospitals [18].

These problems may be solved by a well-designed process flanked by adequate incentives. However, little empirical evidence is available regarding the process to transform quality improvement programs into societal cost savings [2]. Our aim was to contribute to this gap by identifying the barriers and facilitators within the process. We used the implementation of prehabilitation in a university hospital in the Netherlands as a test case. Prehabilitation is a preoperative multimodal lifestyle improvement program for patients undergoing major surgery. Research has shown that prehabilitation could reduce the number of surgical complications, reoperations and the average length of stay [19–21]. Moreover, a recent systematic review of economic evaluations revealed evidence that prehabilitation can be cost-effective compared to usual care [22]. However, these evaluations lack a comprehensive perspective on the costs and savings [22].

## Methods

### Study design and scope

In this study, we conducted semi-structured interviews with relevant stakeholders of prehabilitation in an academic hospital in the Netherlands. Converting freed hospital capacity into societal cost savings is a multi-step process. Our objective was to identify barriers and facilitators associated with these steps. We considered reduced health insurers' costs as the main mechanism to obtain societal savings, given the non-profit structure and public financing of health insurers in the Netherlands [16]. In June 2023, the local medical ethics review committee of the Radboud University Medical Center waived the review of this study as the Medical Research involving Human Subjects Act did not apply (file number: 2023–16520). The Consolidated Criteria for Reporting Qualitative Research (COREQ) were followed and the completed checklist can be found as S1 File [23].

### The test case

Prehabilitation is an important and well-known quality improvement initiative in the hospital. It was gradually implemented between 2021 and 2023 for all high-impact surgery care pathways in seven departments. The intervention consisted of an exercise program, dietetic consultation, psychological support, and smoking cessation support. Prehabilitation has shown positive results on the number of surgical complications, reoperations and the average length of stay [19–21]. Its effectiveness is currently investigated in large scale studies. The hospital financed the implementation and prefinanced the intervention costs. The hospital agreed on a shared-savings agreement with health insurers, anticipating that after about five years after implementation the financial value of the freed capacity would compensate the investment of both health insurers and the hospital.

### Setting

In the Netherlands, hospitals compete for contracts with insurers [24]. While there are ten health insurers in 2024, the four dominant insurers collectively hold approximately 90% of the market share, with variations in market shares across regions [25]. The majority of the hospitals are reimbursed through a hospital DRG-like (Diagnosis Related Group) system called DBCs (Diagnose-Behandel-Combinatie, or Diagnosis Treatment Combination) [24]. Many insurers institute a global budgetary limit, either as lump-sum global budget or claims cost ceiling [26]. In the concerning hospital, the vast majority of the medical staff and employees are salaried on a fixed working hours contract.

## Recruitment and sampling strategy

The stakeholders were recruited via purposive sampling based on experience, current position and department, and affinity with prehabilitation. While using expert sampling, we aimed to include experienced stakeholders from all relevant clinical and facilitating departments, as well as health insurers. We considered hospital managers to be experts, therefore we invited all hospital managers of the involved clinical departments: surgery, intensive care units and operation rooms. Of the facilitating departments (hospital sales, care administration, business administration), we invited employees who were consulted for the internal prehabilitation business case. We invited two persons working for different insurers. Both were involved in the implementation of prehabilitation. After each interview, the participants were asked to suggest stakeholders they deemed relevant for this study. The suggested persons were also invited. All stakeholders were invited per e-mail to participate in a semi-structured interview between the 13th of June 2023 and the 2nd of November 2023. A reminder was sent in case of no response after three to four weeks.

## Data collection

The interviews took place between July 4th and November 22nd 2023. All participants provided verbal informed consent prior to the start of the interview, which was also recorded. Fifteen interviews were conducted by two female researchers DK (MD and MSc) and SvD (PhD), four interviews solo by DK, and one by DK and PJ (male, PhD). All interviewers have experience with qualitative research methods and were not previously involved in the prehabilitation program. The participants knew about their backgrounds and were aware of the study design and objectives. There was no prior relationship between the interviewers and the participants, other than that most participants worked for the same hospital as the interviewers.

The topic guide can be found as S2 File. DK preformed an unstructured literature search to identify possible steps in the process of capturing societal savings and potential barriers and facilitators. DK, NS, SvD and PJ discussed the literature, and shared knowledge and experiences. During iterative meetings, Dk, NS, SvD and PJ identified four stages in the process of capturing societal savings: 1. Reducing capacity, 2. Reducing departmental expenses, 3. Reducing hospital expenses, 4. Reducing insurer costs. These stages were extracted from literature and represent a possible pathway towards societal cost savings [3, 9–11, 13, 27]. DK drafted a interview guide based on the topics discussed during the meetings. The interview guide was reviewed by five team members (SvD, NS, TK, GW and PJ) and adapted based on their feedback. The topic guide was pilot tested with two prehabilitation program managers and a few questions were added to the topic guide. The two pilot interviews were also included in the analysis. The topic guide was slightly adapted for each stakeholder to fit the stakeholders' experience and expertise. Additionally, after each interview, the topic guide was evaluated and extended when the interviewees mentioned new perspectives.

The interviews were preferably held during a face-to-face meeting. If that was inconvenient, the interviews were conducted via a video call using Microsoft Teams. Only the interviewers and participants were present during the interviews. Field notes were made during the interviews to direct further questions. Data saturation was defined as the point in coding when no new barriers or facilitators were identified in two subsequent transcripts. No new interviews were conducted after data saturation was reached. No repeat interviews were carried out and the transcripts were not returned for correction to the participants.

## Data analysis

The interviews were audio-recorded during face-to-face meetings and video-recorded during video-calls. The recordings were transcribed ad verbatim and the transcripts were analysed in

ATLAS.ti. Two authors (DK and SvD) performed a inductive content analysis using the constant comparative method. DK and SvD independently coded the first transcripts. Open coding was used to label barriers and facilitators. If participants mentioned that the presence of an influencing factor hindered the process, it was labelled as a barrier. If the presence would facilitate the process, it was labelled as a facilitator. The barriers and facilitators were categorized into the four stages: reducing capacity, reducing departmental spending, reducing hospital expenses, reducing insurer expenses. During the coding, it came apparent that the first stage consisted of multiple steps. DK and SvD inductively created substages to emphasize these required steps. This was discussed during multiple meetings. Agreement on the coding was reached after five transcripts and the other transcripts were coded by one author (DK) and checked by another (SvD). Differences were resolved in consensus meetings with DK and SvD. The results were discussed during a meetings with NS and PJ.

## Results

We interviewed 20 stakeholders, 10 male and 10 female, of which the functions can be found in Table 1. Three invited professionals did not participate: one surgeon did not reply, one surgeon rejected due to lack of affinity with the subject, and one hospital sales manager rejected due to lack of time. Nine interviews were held face-to-face, and 11 interviews via Microsoft Teams. The duration of the interviews ranged from 27 to 62 minutes.

From 20 interviews, we identified 20 barriers and 23 facilitators across the four stages: reducing capacity, reducing department spending, reducing hospital spending, reducing insurer spending. These can be found in Table 2. Each stage of the process was considered a prerequisite for advancing to the subsequent stage, all aimed at achieving societal cost savings by reducing insurer expenses. The first stage, reducing capacity, has three subcategories: creating excess capacity, preventing substitution with other care, and downsizing.

**Table 1. Function of participants and their relation with prehabilition.**

| Function | Number* | Relation with prehabilitation |
|---|---|---|
| **Medical doctor** | 6 | Three were treating patients after prehabilitation |
| | | Three were selected due to their second position |
| **Hospital manager** | 4 | All were responsible for the finances of departments that were affected by prehabilitation (surgery, operation rooms and intensive care units) |
| **Program manager** | 3 | Two were active prehabilitation program managers |
| | | One was a former prehabilitation program manager and active program manager of other quality improvement initiatives |
| **Hospital administrator** | 3 | Two were consulted during the development of business case of prehabilitation, one was the financial advisor of a relevant department |
| **Health insurer** | 2 | Both were involved in the prehabilitation case |
| **Nurse** | 2 | Both nursed hospitalized patients after prehabilitation |
| **Business controller** | 1 | Was consulted during the development of business case of prehabilitation and during the implementation of prehabilitation |
| **Hospital sales manager** | 1 | Was involved in making agreements concerning prehabilitation |
| **Internal strategy consultant** | 1 | Was familiar with prehabilitation, no direct involvement with prehabilitation, has experience with other quality improvement initiatives |
| **Hospital board member** | 1 | Was responsible for the financing of prehabilitation and external agreements |

*20 respondents were interviewed, some participants have multiple functions, e.g., medical doctor and manager.

**Table 2. Barriers and facilitators per stage of the process of translating capacity savings into societal cost savings.**

| Stage | Barrier | Facilitator |
|---|---|---|
| **1. Reducing capacity** | **Creating excess capacity**<br>• The perceived freed up capacity may be lower than the calculated capacity reductions<br>**Preventing substitution with other care**<br>• Demand for care exceeds existing supply<br>• Substitution is highly valued<br>• Supplier-induced demand allows substitution with other care<br>• Financial incentives stimulate substitution<br>• Non-financial incentives stimulate substitution<br>**Downsizing**<br>• Hospital employees have an aversion towards downsizing<br>• Minimum capacity constraints prevent downsizing<br>• A high threshold must be reached before excess capacity can be scaled down<br>• The hospital board has a restrained approach towards downsizing<br>• Hospital employees have a preference to use excess capacity instead of downsizing | • Combining multiple initiatives allows reaching capacity reduction thresholds<br>• Active management could prevent substitution<br>• The drive to provide appropriate care could counter supplier-induced demand<br>• Insurance agreements can counter unwarranted financial incentives<br>• Optimizing collaboration within the hospital can reduce the minimum capacity<br>• Securing the department's income counters the fear of losing revenue<br>• A strategy could facilitate downsizing<br>• Labor market issues may lead to downsizing |
| **2. Reducing departmental expenses** | • Providing less care does not automatically lower departmental expenses<br>• A high threshold most be reached before working hours can be reduced<br>• Reducing expenses requires time<br>• Insurance agreements do not directly impact the department's decision making | • The hospital board commits to achieving cost savings<br>• A top-down cost-cutting strategy could reduce department budgets<br>• Flexible staffing allows reductions in working hours<br>• High turnover of personnel allows reductions in working hours<br>• Trust that any savings will be well-utilized can motivate employees to reduce expenses<br>• The department benefits from realizing savings, i.e., as part of shared-savings agreements |
| **3. Reducing hospital expenses** | • Reduced departmental expenses do not automatically reduce the hospital expenses<br>• Hospitals have a reluctance to effectuate savings<br>• The diversity of insurance agreements may misalign cost-cutting incentives | • Hospitals have authority to enforce budgetary constraints<br>• A long-term cost-cutting strategy is needed for reducing hospital expenses<br>• Securing the hospital's income counters the fear of losing revenue<br>• Aligned agreements with all involved insurers could prevent free-rider behavior |
| **4. Reducing insurer expenses** | • DBC rates do not automatically decline when an initiative is effective<br>• Agreements on the total hospital budget hinder transfers of cost savings | • The hospital's belief that savings need to be returned to society via insurers<br>• There is a common responsibility to maintain affordable healthcare<br>• Some agreements could to transfer savings from the hospital to insurer<br>• Shared savings agreements could motivate hospitals to reduce costs<br>• Scaling initiatives to other hospitals may increase societal cost savings |

DBC: Diagnosis Treatment Combination, the Dutch equivalent of Diagnosis Related Groups (DRGs)

## Stage 1. Reducing capacity

**Creating excess capacity.** The starting assumption is that a decrease in length of stay reduces the required hospital capacity. Some interviewees agreed with this, while others have questioned whether the nurses' workload decreases proportionally. In particular in the ICU, the first admission day has a higher workload than the following days. Reducing the length of stay may therefore have less impact on the capacity than preventing the admission. In addition, tasks like training patients to self-care and lifestyle adjustments, still needs to be performed before the hospital discharge. Consequently, the perceived excess capacity may be lower than presumed. On the other hand, excess capacity can be enlarged by implementing multiple initiatives that reduce length of stay.

**Preventing substitution with other care.** To reduce the created excess capacity, it is essential that any reductions in care are not filled with other care. However, most participants expect that the excess capacity will be used for other patients. It is frequently mentioned that the demand of care currently exceeds the available supply, and it is expected that the demand will further increase in the future. The interviewees emphasized that some specialized health-care professionals are scarce and should therefore be deployed most efficiently. They considered the opportunity to provide more care with the same resources highly valuable.

*Quote: 'You have a whole operating room complex with all kinds of people ready to do various things. It is wasteful, also a societal waste, not to deploy those people effectively. So, you should let them operate as efficiently as possible. As long as there is demand, as long as there are waiting lists. But that should not be the basis of treatment decisions. It is more like: if people are already on the waiting list, then you want to help as many as possible.'* Participant 13, hospital sales manager

Interviewees expect additional supplier-induced demand as a consequence of available excess capacity. For example, indications for treatment may expand when capacity becomes available. Additionally, the presence of excess capacity reduces the pressure to discharge patients, which may lead to prolonged length of stay of other admitted patients. Participants perceive financial and non-financial incentives to provide care. For example, specialists need to reach target volumes to preserve their competence. On the other hand, providing less care is discouraged, because one may lose opportunities for research, their status, their patients, and departments may need to downsize their capacity. Furthermore, not using full capacity may conflict with other process indicators on which the departments are assessed, such as warm-bed time.

*Quote: 'If [the board] would say: "The ICU is now five million short, and we must reduce staff or whatever", that would be the most foolish thing there is. And then I am going to admit patients to the ICU, who do not belong there, for 3,500 euros. I can earn my money if I want to. I can earn it easily. What all ICUs are doing now is admitting their Medium Care patients to the ICU and billing them as ICU beds. That is what is happening in the Netherlands now.'* Participant 4, medical doctor

The stakeholders also mentioned facilitators. Some participants deemed it possible to prevent substitution with active management and a top-down approach. Furthermore, the drive to provide appropriate care and prevent inappropriate care could counter supplier-induced demand. For example, patients do not always benefit from additional care, especially in the case of ICU treatment. Moreover, the demand for intensive care is decreasing, further reducing substitution possibilities. Lastly, increasing care provision may not be profitable for the hospital if the insurer instated a budgetary cap.

**Downsizing hospital capacity.** To render cost savings, the hospital should minimize its expenses. A viable approach involves downsizing the departmental excess capacity. Participants have expressed negative associations with downsizing in general, and they offered barriers specifically for downsizing excess capacity.

The interviewees expressed an aversion towards downsizing in general. Downsizing is a sensitive matter, and the culture within an organisation and the behaviour of individuals can hinder the process. There are negative perceptions of downsizing, such as it being the start of a slippery slope. Participants stated that once you shrink, you will not be able to retrieve the capacity in the future. Downsizing is also associated with the risk of losing expertise, status, the

market position of the departments and the hospital. Moreover, participants fear losing flexibility in providing care, and consequently foresee increasing problems with the planning and coordination.

> Quote: 'I would absolutely not be in favor of reducing eighteen beds by two or four. Soon, you will have nothing left, and I see it happening now at [department]. In the past, I had ten, twelve beds, and we could provide excellent service to the region. I have now been reduced to six. It is a disaster; it is a disaster to schedule your surgeries, and you have no flexibility anymore. But you are also nothing. You become almost a joke in the region. We need to create a [specialty] network now and we are bringing a six-bed facility. Honestly, I am ashamed.' Participant 3, hospital manager

Participants also mentioned barriers for reducing the excess capacity. First, the presence of excess capacity does not automatically mean that it can be reduced. A reduction in capacity typically requires reaching a certain threshold. Moreover, the downsizing potential is limited by factors such as the requirement of minimal staffing and the need for resilience in case of outbreaks or disasters. Moreover, various participants stated that the departments are already at a minimal capacity. In their perception, they cannot shrink any further, for example because of the need to meet volume norms and retain income. Not meeting these will have negative consequences for departments, such as a loss of revenue. Additionally, tertiary medical centres have certain responsibilities, such as unlimited accessibility for patients in need of tertiary care. If they fail, they risk losing their credibility. Moreover, some participants believe that excess capacity should not be downsized, but the available time should be invested in quality enhancing tasks, such as innovating and teaching. And last, according to participants, the hospital board and the government currently do not make the necessary decisions in reallocating resources and do not steer towards downsizing.

Downsizing is facilitated by improved collaboration within the organization and within the region. This could for example reduce the minimal required staffing. Additionally, a top-down downsizing strategy, endorsed by healthcare professionals, could facilitate the process. This could involve for example establishing explicit agreements and incorporating follow-up mechanisms and data. Furthermore, there is an increasing shortage of nurses and operation room employees. If resigned staff cannot be replaced due to these shortages, downsizing is inevitable. Last, it is suggested that lump-sum payment agreements could overcome the barrier of risking loss of revenue.

**Stage 2. Reducing departmental expenses.** Some participants stated that providing less care does not naturally lead to lower departmental expenditures. The costs saved directly are the variable costs, i.e., material costs. Some interviewees estimated that such costs are only a small percentage of the total departmental costs. The semi-variable costs, e.g., personnel costs, can eventually be reduced, but require reaching a substantial reduction in care. The participants mentioned that this may be difficult due to the small patient numbers in their hospital. In addition, it takes time before expenses can be lowered. The interviewees expect that healthcare professionals will substitute freed-up time with other valuable tasks before the threshold to permanently close a single bed is reached. Such substitution could reduce the perceived quantity of excess capacity. Furthermore, the interviewees mentioned the department's fixed costs, which cannot be reduced easily. As previously mentioned, departments require minimal staffing, for example to cover all shifts and to ensure quality and safety during the shifts. A participant estimated that the fixed costs alone already exceed 50% of total costs. In addition, clinical departments finance supporting departments, such as the operating rooms and the radiology department. Departments can reduce the number of required services, but this does

not substantially reduce the expenses of the supporting departments. To avoid a negative balance, either the fees per services must increase or the free capacity must be used for providing other care. Some participants do not expect that departments will voluntarily reduce expenses. Therefore, a top-down approach may be necessary.

*Quote: 'If you really want to cut costs, then, of course, you have to do fewer things on a large volume. That is always the pain point in an academic medical center [. . .]. And for us, because we have small volumes, does it mean I have to remove a nurse's arm or a leg? Well, that often just does not work.'* Participant 12, hospital board member

A few participants also pointed out that healthcare is a complex system, and that a change in one place can also have consequences somewhere else. For example, reducing spending on staffing can result in less flexibility in care delivery, and it can subsequently induce workload and extra costs for the coordination of care. Lastly, departments and healthcare professionals lack awareness about existing agreements with insurers, limiting the impact of these agreements.

*Quote: 'What I often see is that various initiatives are penny-wise and pound-foolish. So, we save fifteen cents with a specific procurement action, but then we do not realize that suddenly we have additional costs because we have added another provider with whom contracts are made, so someone else incurs those extra costs. [. . .] But also, for example, those five nurses that had been cut back [. . .], you can present that as a significant saving. I am afraid that it has also resulted in us not achieving the revenues because we simply could not accommodate the patients.'* Participant 7, hospital administrator

There are also facilitators to lower departmental expenditures. The hospital board commits to achieve cost-savings with the implementation of prehabilitation, making the deployment of a cost-cutting strategy more likely. Moreover, reducing costs can be rewarded by shared-savings agreements that return part of the savings to the department. To stimulate capacity reductions, excess capacity in the ward and in the operating rooms can be adopted by other specialties. For example, IC nurses can work on the emergency department or the coronary care unit (CCU). In addition, workforce reductions could be achieved by phasing out through natural outflow due to a high turnover of nurses, rather than resorting to terminations. Furthermore, there is currently a shortage of nurses, causing understaffing. Apart from the potential negative consequences, this could also reduce departmental expenses. Another driver to reduce departmental spending is the trust that savings are purposefully spent. A participant stated that the value for money may increase when financial resources are reallocated towards other sectors, for example elderly care. Excessive spending on a few patients could be perceived as incompatible with budgetary constraints elsewhere. If the stakeholders trust that savings would be spent wisely, it could enhance their motivation to reduce costs.

**Stage 3. Reducing hospital expenses.** Reducing departmental expenses does not necessarily reduce the total hospital costs. For example, when excess capacity is absorbed by other departments, e.g., when a nurse works in a different department, the total hospital spending remains unchanged. Participants stated that a multi-year plan is needed to effectuate the cost-savings. Another participant mentioned that the hospital board often does not specifically enforce case-based cost savings, while that is perceived as necessary.

*Quote: 'We chose to implement prehabilitation. That simply means that you have excess capacity in other areas. So, in those areas, you also need to achieve your savings. And if not, if*

*you are not willing to do that, then you also need to have the courage to say: we do not want this and we stop offering prehabilitation. That is also an option'* Participant 9, strategy consultant

Dutch hospitals negotiate with multiple health insurers, resulting in various budgetary agreements. A participant mentioned that misalignment of incentives for downsizing and cost-cutting may consequently occur. For example, one agreement may consist of a lump-sum payment, securing the hospital's income while reducing excess capacity, while a cost ceiling agreement with a different insurer could reduce hospital income when excess capacity is reduced. In addition, participants deemed transferring departmental savings via the hospital to the insurers as complex. First, there is a lack of insight in how costs are structured, hindering monitoring actual cost-savings. Besides, some interviewees stated that there is a misalignment between internal budgeting and reimbursement, further complicating the transfer.

*Quote: 'Because from a specialization perspective, you focus on your production plan, and as a hospital, you focus on the required Full-Time Equivalents (FTEs), but there is an indirect link in that. It is not a one-to-one relationship. Additionally, you also deal with a whole cost price system. [. . .] So the question is: Are the savings of those few FTEs sufficiently reflected in the system that lies underneath it? Often, it is just rounded off.'* Participant 15, hospital administrator

Mentioned facilitators include a secured hospital income and aligned agreements with the involved insurers. This would enable the hospital to reduce expenses without the fear of incurring a loss. Additionally, the stakeholders emphasized the role of the hospital board. The board has the authority to enforce departmental budgetary constraints to reduce the hospital expenditure. In this scenario, the individual departments retain the authority to determine cost-cutting measures, which may not necessarily involve reducing excess capacity.

**Stage 4. Reducing insurer expenditures.**   Several participants stated that savings should be effectuated by the health insurers through lower premiums. In addition, a stakeholder deemed achieving hospital savings a prerequisite to transfer any savings to the insurers. Reduction of insurers' costs largely depends on the agreements between the hospital and the health insurers. Participants named reducing the DBC (the Dutch equivalent of Diagnosis Related Groups (DRGs)) rates or reducing the number of reimbursed DBCs as a way to transfer savings to society. However, they also stated that the DBC rates are not aligned with actual hospital costs and health insurers lack insight in the hospital expenditures, therefore DBC rates do not automatically decline when an initiative is effective. Therefore, cost savings may depend on specific agreements with hospitals. These are, however, often lacking because the savings potential of a single quality improvement initiative may face too many transaction costs to be included in the negotiation process. Another participant stated that reducing the DBC rates or the number of DBCs does not automatically reduce insurer costs. Agreements are made on the level of both the DBC rate and the total hospital budget. Budgetary caps could hinder translating hospital savings to the insurer, because insurers may not have to reimburse the full costs or not reimburse at all when a cap is reached. Therefore, savings on the level of DBCs are not automatically transferred to the payers. Also, in case of lump sum agreements, lowering of rates or the number of DBCs does not influence the reimbursed amount.

Some participants also mentioned some agreements between hospitals and insurers that facilitate the transfer of savings. For example, an open-ended budget automatically reduces hospital expenditures in case of volume reductions or reductions in the DBC rate. Also shared-savings agreements could transfer part of the hospital savings, while additionally

motivate stakeholders to reduce costs. However, within a shared savings model, participants wonder how much will be left when the savings are shared with all stakeholders. Moreover, interviewees of the hospital and an insurer mentioned free-rider behavior by other insurers. Therefore, aligned agreements of insurers is mentioned as a prerequisite. Another proposed solution is a multiyear agreement, because it can provide the hospital time to reduce their cost structure. However, participants mentioned that insurers are reluctant on such agreements, for example because of the uncertainties of price fluctuations. Last, innovation-specific agreements are also mentioned to transfer hospital savings to insurers.

> Quote: 'I personally find the shared savings model to be a good principle because, ultimately, they are societal costs. Or it has been contributed by society, so it should flow back in that direction. But, of course, it is quite elegant if the hospital benefits from it as well. I mean, that is just where it starts. Every change process is simply individuals asking themselves, 'What's in it for me?'' Participant 9, strategy consultant

Another facilitator is the common goal of stakeholders to keep healthcare affordable and innovative. It is perceived to be a societal responsibility to contain the healthcare spending. In addition, investments of insurers may lead to external pressure for hospitals to effectuate savings. And last, scaling the innovation to more hospitals is also seen as a way to enhance societal savings.

## Discussion

Twenty barriers and 23 facilitators were identified in four stages to capture societal cost savings: reducing capacity, department expenses, hospital expenses and insurer expenses. In general, there is an aversion towards downsizing. Due to lack of incentives to reduce costs or top-down policies for downsizing, all participants expect that any excess capacity will be used to provide other care. Nevertheless, such substitution is perceived as valuable and a societal gain. Other mentioned barriers are fear of losing resilience, flexibility, status and revenue. Moreover, agreements with a budgetary cap and lump sum agreements may hinder the translation of the cost savings to the insurers. And last, misalignment of agreements between hospitals and health insurers creates financial barriers for downsizing and cost-cutting. Identified facilitators included shared savings agreements, a downsizing strategy, labor shortages, and a shared responsibility to secure affordable healthcare, among others.

The identified barriers indicate that monetizing savings for society does not occur automatically when an initiative is effective. Stakeholders expect that saved capacity will be used to provide other care. This aligns with existing literature describing supplier-induced demand in healthcare [14, 15, 28, 29]. Reducing length of stay only saves a small percentage of the expenditure directly, because personnel costs and fixed costs remain unaffected in the short term [10]. A study illustrated that a reduction of 12 beds, typical for a ward, enables personnel reorganization and substantially reduce semi-fixed costs [27]. However, such large-scale downsizing requires a large volume of excess capacity. This may require combining multiple quality improvement initiatives as part of a hospital-wide strategy [9]. Moreover, stakeholders mentioned the need for a strategy or active approach to achieve reductions on all four stages. For example, a strategy is also deemed necessary to subsequently transfer the hospital savings to insurers due to existing misaligned agreements and incompatible budgeting systems [13, 30].

Another identified facilitator is to secure the departments and hospital's income. However, securing either the departments or hospital's income and achieving societal costs savings seem incompatible. Nevertheless, this may be possible when savings are interpreted as a reduction in hospital spending growth rate compared to a historical benchmark. Shared-savings

agreements between parties may accommodate this, although past experiments yield mixed results [9, 31, 32].

Downsizing is deemed controversial by participants. The stakeholders emphasize that the demand for care currently exceeds the supply and it is expected to further increase [33]. Substituting excess capacity with other necessary care may partly compensate for increasing demand. Therefore, providing more care with approximately the same resources could also be considered as a societal gain of quality improvement initiatives. In addition, it may not be necessary to aim for downsizing, since increasing shortages of medical professionals may cause natural downsizing in the future [33, 34]. In this case, effective prehabilitation could offer the opportunity to increase efficiency, and thereby retaining the accessibility of care.

Healthcare decision-making may be improved by broadening the scope of the value of quality improvement initiatives [35]. The value of quality improvement initiatives may cover a broader range than cash savings and saved hospital capacity [36]. For example, prehabilitation may additionally reduce home care and could lead to earlier return to work [20]. Furthermore, the identified barriers suggest that monetizing capacity savings is difficult and that the saved amount may be lower than expected. This study suggests that the value of reducing length of stay is to be able to provide care for other patients. Therefore, only expressing the value in terms of costs saved lacks important nuances. By also focusing on the effects on the increased accessibility, healthcare decision making may be improved. Future research should focus on the value of care substitution and the impact of care substitution on the accessibility.

## Strengths and limitations

To our knowledge, this is the first study that identifies barriers and facilitators through the entire process from an effective quality improvement initiative towards reducing societal costs. Additionally, a broad range of relevant stakeholders participated in this study. Some limitations apply. First, even though studies on prehabilitation show promising results, the effectiveness of our test case was yet unknown during the interview period. Consequently, certain questions were framed hypothetical, e.g., 'what if . . .'. To substantiate expectations, participants were additionally asked for examples and experiences with other quality improvement initiatives. Secondly, with the exception of two insurer employees, all stakeholders were affiliated with the same hospital. Therefore, some barriers and facilitators may be context specific. Last, this article solely focuses on achieving societal cost savings through the described four stages and does not address the conversion of reduced health insurers' costs in societal savings in the form of lower premiums or governmental expenses. Nor does this article address alternative ways quality improvement initiatives could generate societal savings.

## Conclusion

This study describes barriers and facilitators in the process of capturing societal cost savings across four stages: 1) reducing capacity, 2) reducing department expenses, 3) reducing hospital expenses, and 4) reducing insurer expenses. An encompassing hospital strategy targeting these four stages is recommended, because societal cost savings do not occur automatically when hospital capacity is saved. Shared-savings agreements could facilitate the transfer of hospital cost savings to the health insurers. However, many barriers were encountered. Predominantly, stakeholders expect that any saved capacity will be used for other care due to increasing demand. However, such substitution with other care is also perceived as a societal gain. Framing financial gains of quality improvement initiatives in terms of addressing increasing demand may therefore be more accurate. This would require additional research into the value of care substitution.

## Supporting information

**S1 File. Completed checklist of the Consolidated Criteria for Reporting Qualitative Research (COREQ).**
(PDF)

**S2 File. Topic guide.**
(DOCX)

## Acknowledgments

We would like to thank the interviewees for their time and effort.

## Author Contributions

**Conceptualization:** Daniëlle Kroon, Simone A. van Dulmen, Niek W. Stadhouders, Patrick P. T. Jeurissen.

**Formal analysis:** Daniëlle Kroon, Simone A. van Dulmen.

**Funding acquisition:** Patrick P. T. Jeurissen.

**Methodology:** Daniëlle Kroon, Simone A. van Dulmen, Niek W. Stadhouders.

**Supervision:** Jonas Rosenstok, Baukje van den Heuvel, Gert P. Westert, Rudolf B. Kool, Patrick P. T. Jeurissen.

**Writing – original draft:** Daniëlle Kroon.

**Writing – review & editing:** Daniëlle Kroon, Simone A. van Dulmen, Niek W. Stadhouders, Jonas Rosenstok, Baukje van den Heuvel, Gert P. Westert, Rudolf B. Kool, Patrick P. T. Jeurissen.

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
