## [Decision Letter · Decision Letter 0]

17 Jul 2024

PONE-D-24-04936Stakeholders’ perspectives on capturing societal cost savings from quality improvement initiatives: a qualitative studyPLOS ONE

Dear Dr. Kroon,

Thank you for submitting your manuscript to PLOS ONE. After careful consideration, we feel that it has merit but does not fully meet PLOS ONE’s publication criteria as it currently stands. Therefore, we invite you to submit a revised version of the manuscript that addresses the points raised during the review process.

We look forward to receiving your revised manuscript.

Kind regards,

Cigdem Kadaifci, Assoc. Prof.

Academic Editor

PLOS ONE

“The Dutch Ministry of Health, Welfare and Sport (grant number 331032).”

“JR and BvdH are program leaders of the prehabilitation implementation. Because they are also important stakeholders, they both participated as interviewee. The other authors have no competing interests to declare.”

Reviewers' comments:

Reviewer's Responses to Questions

**Comments to the Author**

1. Is the manuscript technically sound, and do the data support the conclusions?

Reviewer #1: Yes

Reviewer #2: Yes

2. Has the statistical analysis been performed appropriately and rigorously? 

Reviewer #1: N/A

Reviewer #2: N/A

3. Have the authors made all data underlying the findings in their manuscript fully available?

Reviewer #1: No

Reviewer #2: Yes

4. Is the manuscript presented in an intelligible fashion and written in standard English?

Reviewer #1: Yes

Reviewer #2: Yes

5. Review Comments to the Author

Reviewer #1: Thank you for the opportunity to peer-review the manuscript “Stakeholders’ Perspectives On Capturing Societal Cost Savings From A Quality Improvement Initiative: A Qualitative Study “ by Kroon et al .I congratulate the authors on this well-performed and very interesting study. I believe that their paper will be very stimulating for the discussions about implementing prehabilitation pathways internationally!

Before accepting this manuscript for publication, I would recommend addressing the minor points below:

Abstract: The results should be written in past tense.

Methods - Study design and scope: Please provide the date of the ethics waiver (at least month and year, so that it becomes apparent if that was sorted before the first interview).

Methods - Recruitment and sampling strategy:

Suggest introducing a definition of data saturation here and state that you would stop once it was reached. Please consider moving the following information to the results part: “We interviewed 20 stakeholders, 10 male and 10 female, of which the functions can be found in table 1. Three invited professionals did not participate: one surgeon did not reply, one surgeon rejected due to lack of affinity with the subject, and one hospital sales manager rejected due to lack of time.”

Table 1: Suggest to sort by number, suggest to move to results.

Methods – Data collection:

Please consider moving the following information to the results part: “Nine interviews were held face-to-face, and 11 interviews via Microsoft Teams. The duration of the interviews ranged from 27 to 62 minutes. [….] Data saturation was reached after 20 interviews.”

Was the interview guide pilot tested?

Methods - Recruitment and sampling strategy: Suggest to introduce a definition of data saturation here and state that you would stop once it was reached.

Methods – Data analysis: Did you follow specific transcribing rules, e.g. regarding “ehms” and things like laughter etc.?

Which methodological orientation and theory did you follow? How did you create the coding tree? I.e. how did you create the three subcategories of the first category, how did you decide what constitutes a barrier and what constitutes a facilitator?

Results:

Table 2: Write out “DBC” or provide abbreviation in a legend.

Quote in l. 203: Consider translating it with “… then you want to help as many as possible.” when referring to people.

Discussion: societal cost savings is more than just reducing insurer expenses in my view

Appendix:

COREQ checklist: Provide page numbers please. Also, please re-consider if items 6, 8, 15, 18 and 28 really were not applicable or whether you did not do them. The later is fine but you should be transparent about it, e.g., you should state that no relationship established prior to study commencement (item 6), if that was not the case.

Data sharing statement: Please add a statement why transcripts cannot be shared.

On a side-note: In Germany, we are facing similar issues like the ones covered in your study from the Netherlands. It will be interesting what the ongoing hospital reform in Germany (Hospital Remuneration Act) will change. E.g., the plan is to add retention sum lumps to the current system of per-case flat rates, so that hospitals receive money for offering certain services (not per service provided). Downsizing (and closure of hospitals) is a very emotional/ and mostly negatively depicted topic among the public (https://link.springer.com/article/10.1007/s10389-024-02271-6). However, many citizens don’t know that hospitals are already forced to close down wards due to lack of nursing staff (which you discussed also).

Best wishes,

Tanja Rombey

Reviewer #2: The authors developed a qualitative study for identifying barriers and facilitators whether quality improvement initiatives such as prehabilitation result in societal cost savings. They conducted semi-structured interviews with stakeholders relevant to implemented prehabilitation program in a Dutch hospital including medical personnel, hospital administrators and health insurers. The questions in the interview are grouped into four categories: reducing capacity, reducing departmental expenses, reducing hospital expenses and reducing insurer costs and the authors identified 20 barriers and 23 facilitators across these categories. The results suggest that quality improvement initiatives that could save capacity may not necessarily result in quantifiable cost-savings due to barriers and increasing demand.

Major comments:

The paper is well-written and it is about an important topic. However, there are some issues I would like to raise about the study.

1) The main assumption of the study is that prehabilitation or other similar quality improvement initiatives can reduce the length of stay in the hospital and increase capacity. Then, this yields to four possible results/categories: reducing capacity, reducing departmental excess, reducing hospital expenses and reducing insurer costs. These categories were selected from a viewpoint article. First, based on the implementation of prehabilitation in this hospital, does it really result with less LOS and increased capacity? Second, if so, have you been able to establish any links between increased capacity and these four possible categories/results? In other words, is there any supporting evidence that links prehabilitation or any other quality improvement initiatives with reducing capacity, departmental excess etc. in your hospital or in the literature? Selection of a viewpoint article for this major assumption seems weak for this study since if this link does not exist, all these discussions would be pointless.

Conversely, any findings here could be applicable to any interventions that yields these results. So, focusing on prehabilitation does not seem necessary which can be seen from the topic guide (very few questions are about prehabilitation). Why did you select of the interviewers with the knowledge of prehabilitation instead of knowledge of quality improvement programs?

2) What type of purposive sampling techniques are used in this study? Are there any similarities and variations among the interview participants? More information about the sampling procedures would be better (e.g. selection criteria, inclusion/exclusion, importance etc.)

3) The authors claim this is the first study that identifies barriers and facilitators through the entire process from an effective quality improvement initiative towards reducing societal costs. Could you provide more literature to support this claim? Are there any other qualitative or quantitative studies (e.g., cost-effectiveness) on this topic (quality improvement initiatives and/or prehabilitation)? Literature in the introduction section could be extended.

4) Thank you for sharing the topic guide. How did you come up with these questions? Could you provide more context on the development of the topic guide?

5) Table 1 presents the function of the participants. It would be better if you could provide more information on the participants, particularly breadth and depth of understanding about the topic of the participants. What is their experience with the prehabilitation and other quality improvement initiatives? More descriptive statistics about the participants would be good.

Minor Comments:

1) COREQ checklist should provide page number.

2) Citation 5 is not available. Unavailable or unpublished work should not be included. Add a citation with arVix or DOI if not published yet.

3) Citation 13 has no journal or DOI information.

4) Title of the citations in foreign languages (e.g., citation 17 and 33) should be translated to English.

5) Page 6 line 133 should be Table 1.

6. PLOS authors have the option to publish the peer review history of their article (what does this mean?). If published, this will include your full peer review and any attached files.

Reviewer #1: **Yes: **Tanja Rombey

Reviewer #2: No

---

## [Author Response · Author response to Decision Letter 0]

6 Aug 2024

Comments to the Author

1. Is the manuscript technically sound, and do the data support the conclusions?

Reviewer #1: Yes

Reviewer #2: Yes

2. Has the statistical analysis been performed appropriately and rigorously? 

Reviewer #1: N/A

Reviewer #2: N/A

3. Have the authors made all data underlying the findings in their manuscript fully available?

Reviewer #1: No

- We have included the reason why the raw transcripts are not publicly shared.

Reviewer #2: Yes

4. Is the manuscript presented in an intelligible fashion and written in standard English?

Reviewer #1: Yes

Reviewer #2: Yes

5. Review Comments to the Author

Reviewer #1: Thank you for the opportunity to peer-review the manuscript “Stakeholders’ Perspectives On Capturing Societal Cost Savings From A Quality Improvement Initiative: A Qualitative Study “ by Kroon et al .I congratulate the authors on this well-performed and very interesting study. I believe that their paper will be very stimulating for the discussions about implementing prehabilitation pathways internationally!

- Thank you for your time and effort to review our manuscript and thank you for your compliments and suggestions.

Before accepting this manuscript for publication, I would recommend addressing the minor points below:

1. Abstract: The results should be written in past tense.

- We have rewritten the result section to the past tense:

P2, L31:‘We identified 20 barriers and 23 facilitators across four stages: reducing capacity, reducing departmental expenses, reducing hospital expenses and reducing insurer expenses. All participants expected that the excess capacity will be used for other priorities. This was perceived as highly valuable and as an efficiency gain. Other barriers to capture savings included the fear of losing resilience, flexibility, status and revenue. Misalignment between service contracts among hospitals and insurers can hinder the ability to financially incentivize cost reductions. Additionally, some contract types can hinder the transfer of hospital savings to insurers. Identified facilitators included shared savings agreements, an explicit strategy targeting all stages, and labor shortage, among others’ 

2. Methods - Study design and scope: Please provide the date of the ethics waiver (at least month and year, so that it becomes apparent if that was sorted before the first interview).

- We have added the following sentence to the method section and provided the date of the first interview: 

P5, L110: ‘In June 2023, the local medical ethics review committee of the Radboud University Medical Center waived the review of this study as the Medical Research involving Human Subjects Act did not apply (file number: 2023-16520).’

P7, L173: ‘The interviews took place between July 4th and November 22nd 2023.’

3. Methods - Recruitment and sampling strategy:

Suggest introducing a definition of data saturation here and state that you would stop once it was reached. Please consider moving the following information to the results part: “We interviewed 20 stakeholders, 10 male and 10 female, of which the functions can be found in table 1. Three invited professionals did not participate: one surgeon did not reply, one surgeon rejected due to lack of affinity with the subject, and one hospital sales manager rejected due to lack of time.”

- We have added a definition of data saturation in the method section: 

P8, L209: ‘Data saturation was defined as the point in coding when no new barriers or facilitators were identified in two subsequent transcripts. No new interviews were conducted after data saturation was reached.’

- We moved both mentioned sentences to the result section on P9, L235.

4. Table 1: Suggest to sort by number, suggest to move to results.

- We have sorted the content of the table by number and moved it to the results section on page 9.

5. Methods – Data collection:

Please consider moving the following information to the results part: “Nine interviews were held face-to-face, and 11 interviews via Microsoft Teams. The duration of the interviews ranged from 27 to 62 minutes. [….] Data saturation was reached after 20 interviews.”

- We have moved the information to the result section on P9, L238. 

6. Was the interview guide pilot tested?

- We added information about pilot testing the interview guide: 

P7, L291: ‘It was pilot tested with two prehabilitation program managers. These interviews were also included in the analysis.’

7. Methods - Recruitment and sampling strategy: Suggest to introduce a definition of data saturation here and state that you would stop once it was reached.

- We have added a definition of data saturation in the method section: 

P8, L209: ‘Data saturation was defined as the point in coding when no new barriers or facilitators were identified in two subsequent transcripts. No new interviews were conducted after data saturation was reached.’

8. Methods – Data analysis: Did you follow specific transcribing rules, e.g. regarding “ehms” and things like laughter etc.?

Which methodological orientation and theory did you follow? How did you create the coding tree? I.e. how did you create the three subcategories of the first category, how did you decide what constitutes a barrier and what constitutes a facilitator?

- We have clarified the used approach in the method section: 

P8, L215: ‘The recordings were transcribed ad verbatim and the transcripts were analysed in ATLAS.ti’

P8, L216: ‘Two authors (DK and SvD) performed a inductive content analysis using the constant comparative method. DK and SvD independently coded the first transcripts. Open coding was used to label barriers and facilitators. If participants mentioned that the presence of an influencing factor hindered a step in the process, it was labeled as a barrier. If the presence would facilitate the process, it was labeled as a facilitator. The barriers and facilitators were categorized into the four stages: reducing capacity, reducing departmental spending, reducing hospital expenses, reducing insurer expenses. During the coding, it came apparent that the first stage consisted of multiple steps. DK and SvD inductively created substages to emphasize these required steps. This was discussed during multiple meetings. Agreement on the coding was reached after five transcripts and the other transcripts were coded by one author (DK) and checked by another (SvD). Differences were resolved in consensus meetings with DK and SvD. The results were discussed during meetings with NS and PJ.’

9. Results:

Table 2: Write out “DBC” or provide abbreviation in a legend.

- We provided an abbreviation in the legend: 

P12, L267 DBC: Diagnosis Treatment Combination, the Dutch equivalent of Diagnosis Related Groups (DRGs)

In addition, we have clarified the abbreviation in the manuscript: 

P5, L133: ‘The majority of the hospitals are reimbursed through a hospital DRG-like (Diagnosis Related Group) system called DBCs (Diagnose-Behandel-Combinatie, or Diagnosis Treatment Combination).’

10. Quote in l. 203: Consider translating it with “… then you want to help as many as possible.” when referring to people.

- We have changed ‘as much as possible’ to ‘as many as possible’.

P13, L294: ‘It is more like: if people are already on the waiting list, then you want to help as many as possible.'

11. Discussion: societal cost savings is more than just reducing insurer expenses in my view

- We agree that reducing societal cost savings may not be the same as reducing insurer costs. 

We have explained our decision to focus on reducing insurer expenses this in the scope: 

Page 5, line 106: ‘Converting freed hospital capacity into societal cost savings is a multi-step process. Our objective was to identify barriers and facilitators associated with these steps. We considered reduced health insurers’ costs as the main mechanism to obtain societal savings, given the non-profit structure and public financing of health insurers in the Netherlands. (16)’

This point was also addressed by the participants and mentioned in the results under the subheading ‘reducing insurer expenditure’ on P19, L453: 

‘Several participants stated that savings should be effectuated by the health insurers through lower premiums.’

We also acknowledged that prehabilitation can add value to society in other ways. We have addressed this in the discussion: 

P23, L533: Healthcare decision-making may be improved by broadening the scope of the value of quality improvement initiatives.(36) The value of quality improvement initiatives may cover a broader range than cash savings and saved hospital capacity.(37) For example, prehabilitation may additionally reduce home care and could lead to earlier return to work.(21)

Additionally, we have emphasized the limitations of our scope in the limitations: 

P24, L554: ‘Last, this article solely focuses on achieving societal cost savings through the described four stages and does not address the conversion of reduced health insurers’ costs in societal savings in the form of lower premiums or governmental expenses. Nor does this article address alternative ways quality improvement initiatives could generate societal savings.’

12. Appendix:

COREQ checklist: Provide page numbers please. Also, please re-consider if items 6, 8, 15, 18 and 28 really were not applicable or whether you did not do them. The later is fine but you should be transparent about it, e.g., you should state that no relationship established prior to study commencement (item 6), if that was not the case.

- Thank you for noting. We reported all items in the manuscript and we inserted the page numbers in the checklist. 

P7, L179: ‘There was no prior relationship between the interviewers and the participants, other than that most participants worked for the same hospital as the interviewers.’

P8, L207: ‘Only the interviewers and participants were present during the interviews.’

P8, L211: ‘No repeat interviews were carried out and the transcripts were not returned for correction to the participants.’

13. Data sharing statement: Please add a statement why transcripts cannot be shared.

- We included an explication for not sharing the transcripts: 

‘Data cannot be shared publicly because the participants did not provide consent to share the transcripts to persons other than the researchers. In addition, the transcripts contain business-sensitive information and even anonymized raw data can be compromising. Data are available from the IQ health science department of the Radboudumc (contact via iqhealth@radboudumc.nl) for researchers who meet the criteria for access to confidential data.’

On a side-note: In Germany, we are facing similar issues like the ones covered in your study from the Netherlands. It will be interesting what the ongoing hospital reform in Germany (Hospital Remuneration Act) will change. E.g., the plan is to add retention sum lumps to the current system of per-case flat rates, so that hospitals receive money for offering certain services (not per service provided). Downsizing (and closure of hospitals) is a very emotional/ and mostly negatively depicted topic among the public (https://link.springer.com/article/10.1007/s10389-024-02271-6). However, many citizens don’t know that hospitals are already forced to close down wards due to lack of nursing staff (which you discussed also).

Best wishes,

Tanja Rombey

Reviewer #2: The authors developed a qualitative study for identifying barriers and facilitators whether quality improvement initiatives such as prehabilitation result in societal cost savings. They conducted semi-structured interviews with stakeholders relevant to implemented prehabilitation program in a Dutch hospital including medical personnel, hospital administrators and health insurers. The questions in the interview are grouped into four categories: reducing capacity, reducing departmental expenses, reducing hospital expenses and reducing insurer costs and the authors identified 20 barriers and 23 facilitators across these categories. The results suggest that quality improvement initiatives that could save capacity may not necessarily result in quantifiable cost-savings due to barriers and increasing demand.

Major comments:

The paper is well-written and it is about an important topic. However, there are some issues I would like to raise about the study.

- Thank you for time and effort to review our manuscript and for the provided feedback and suggestions. 

1) The main assumption of the study is that prehabilitation or other similar quality improvement initiatives can reduce the length of stay in the hospital and increase capacity. Then, this yields to four possible results/categories: reducing capacity, reducing departmental excess, reducing hospital expenses and reducing insurer costs. These categories were selected from a viewpoint article. 

a. First, based on the implementation of prehabilitation in this hospital, does it really result with less LOS and increased capacity? 

- There is empirical evidence that reveals the positive effects of prehabilitation on length of stay and complication rate. We provided additional evidence for this in the introduction: 

P4, L95: ‘Research has shown that prehabilitation could reduce the number of surgical complications, reoperations and the average length of stay.(19-21)’

We agree that it is an assumption that a reduced LOS would automatically mean that there would be excess capacity. Therefore, we asked the participating healthcare professionals: ‘What is the impact of reducing the length of stay on your workload?’ We have added the question to the topic guide. It was previously not included in there because it was only asked to the minority of the participants (only to the healthcare professionals that provided care to prehabilitated patients).

We addressed the relation of reduced length of stay and excess capacity in the results: 

P12, L271: ‘Creating excess capacity

The starting assumption is that a decrease in length of stay reduces the required hospital capacity. Some interviewees agreed with this, while others have questioned whether the nurses’ workload decreases proportionally. In particular in the ICU, the first admission day has a higher workload than the following days. Reducing the length of stay may therefore have less impact on the capacity than preventing the admission. In addition, tasks like training patients to self-care and lifestyle adjustments, still needs to be performed before the hospital discharge. Consequently, the perceived excess capacity may be lower than presumed. On the other hand, excess capacit

---

## [Decision Letter · Decision Letter 1]

8 Sep 2024

Stakeholders’ Perspectives On Capturing Societal Cost Savings From A Quality Improvement Initiative: A Qualitative Study

PONE-D-24-04936R1

Dear Dr. Kroon,

We’re pleased to inform you that your manuscript has been judged scientifically suitable for publication and will be formally accepted for publication once it meets all outstanding technical requirements.

Kind regards,

Cigdem Kadaifci, Assoc. Prof.

Academic Editor

PLOS ONE

Additional Editor Comments (optional):

Reviewers' comments:

Reviewer's Responses to Questions

**Comments to the Author**

1. If the authors have adequately addressed your comments raised in a previous round of review and you feel that this manuscript is now acceptable for publication, you may indicate that here to bypass the “Comments to the Author” section, enter your conflict of interest statement in the “Confidential to Editor” section, and submit your "Accept" recommendation.

Reviewer #1: All comments have been addressed

Reviewer #2: All comments have been addressed

2. Is the manuscript technically sound, and do the data support the conclusions?

Reviewer #1: Yes

Reviewer #2: Yes

3. Has the statistical analysis been performed appropriately and rigorously? 

Reviewer #1: N/A

Reviewer #2: N/A

4. Have the authors made all data underlying the findings in their manuscript fully available?

Reviewer #1: Yes

Reviewer #2: Yes

5. Is the manuscript presented in an intelligible fashion and written in standard English?

Reviewer #1: Yes

Reviewer #2: Yes

6. Review Comments to the Author

Reviewer #1: The author's have satisfactorily addressed all my previous comments. I congratulate them on this interesting study!

Reviewer #2: (No Response)

7. PLOS authors have the option to publish the peer review history of their article (what does this mean?). If published, this will include your full peer review and any attached files.

Reviewer #1: **Yes: **Tanja Rombey

Reviewer #2: **Yes: **Emine Yaylali

---

## [Editor Report · Acceptance letter]

13 Sep 2024

PONE-D-24-04936R1 

PLOS ONE

Dear Dr. Kroon, 

I'm pleased to inform you that your manuscript has been deemed suitable for publication in PLOS ONE. Congratulations! Your manuscript is now being handed over to our production team.

Kind regards, 

on behalf of

Dr. Cigdem Kadaifci 

Academic Editor

PLOS ONE